# Calabria as a Genetic Isolate: A Model for the Study of Neurodegenerative Diseases

**DOI:** 10.3390/biomedicines10092288

**Published:** 2022-09-15

**Authors:** Francesco Bruno, Valentina Laganà, Raffaele Di Lorenzo, Amalia C. Bruni, Raffaele Maletta

**Affiliations:** 1Regional Neurogenetic Centre (CRN), Department of Primary Care, ASP Catanzaro, 88046 Lamezia Terme, Italy; 2Association for Neurogenetic Research (ARN), 88046 Lamezia Terme, Italy

**Keywords:** Calabria, Italy, neurodegenerative diseases, Alzheimer’s disease, frontotemporal dementia, Parkinson’s disease, Niemann–Pick type C disease, spinocerebellar ataxia, Creutzfeldt–Jakob disease, Gerstmann–Straussler–Scheinker disease

## Abstract

Although originally multi-ethnic in its structure, nowadays the Calabria region of southern Italy represents an area with low genetic heterogeneity and a high level of consanguinity that allows rare mutations to be maintained due to the founder effect. A complex research methodology—ranging from clinical activity to the genealogical reconstruction of families/populations across the centuries, the creation of databases, and molecular/genetic research—was modelled on the characteristics of the Calabrian population for more than three decades. This methodology allowed the identification of several novel genetic mutations or variants associated with neurodegenerative diseases. In addition, a higher prevalence of several hereditary neurodegenerative diseases has been reported in this population, such as Alzheimer’s disease, frontotemporal dementia, Parkinson’s disease, Niemann–Pick type C disease, spinocerebellar ataxia, Creutzfeldt–Jakob disease, and Gerstmann–Straussler–Scheinker disease. Here, we summarize and discuss the results of research data supporting the view that Calabria could be considered as a genetic isolate and could represent a model, a sort of outdoor laboratory—similar to very few places in the world—useful for the advancement of knowledge on neurodegenerative diseases.

## 1. Introduction

Calabria is a region of southern Italy of 1,839,352 inhabitants [1], which constitutes the tip of the “boot” of Italy. The Strait of Messina separates it from Sicily, and it is suspended between two seas, the Ionian Sea and the Tyrrhenian Sea [2]. This strategic geographical position has favored various migratory flows both in pre-historical and historical times (e.g., Greek, Phoenician, and Carthaginian colonization; Roman, Arab, and Norman conquest) that have determined the presence of different genetic layers in the current population [3,4].

Although originally multi-ethnic in its structure, the Calabrian population has remained relatively stable over the last three centuries, allowing its genetic “imprint” to remain constant over time. This phenomenon was determined by high flows of emigration and low immigration that, together with the geographical characteristics of the region (mountainous with scarce and difficult communication routes), favored the maintenance of closed populations with a high rate of inbreeding, which are the characteristics of genetic isolates. Thus, nowadays Calabria represents an area with low genetic heterogeneity and a high level of consanguinity that allows rare mutations—which originally represent an organism’s response to a better environmental adaptation but become causative of very serious diseases in adults—to be maintained due to the founder effect [5,6]. A typical example is given by the presenilin 1 mutation which has protected over the centuries against very high perinatal mortality (the carriers of the mutation survived) but then develops hereditary Alzheimer’s disease at about the age of 40 years old [5]. In addition, in Calabria there has been reported a higher prevalence of rare autosomal recessive diseases with neurological features—such as fucosidosis [7], hereditary motor and sensory neuropathy [8], benign familial infantile seizures, familial paroxysmal kinesigenic dystonia, and familial infantile convulsions with paroxysmal choreoathetosis [9]—as a genetic consequence of high consanguinity.

Beyond these autosomal recessive diseases and Alzheimer’s disease, other rare and hereditary neurodegenerative diseases have been documented with a high frequency such as frontotemporal dementia, Parkinson’s disease, Niemann–Pick type C disease, spinocerebellar ataxia [10,11,12,13,14], Creutzfeldt–Jakob disease, and Gerstmann–Straussler–Scheinker disease (personal data).

Here, we summarize and discuss the results of research data supporting the view that Calabria could be considered as a genetic isolate and could represent a model, a sort of outdoor laboratory—similar to very few places in the world—useful for the advancement of knowledge on neurodegenerative diseases.

## 2. Neurodegenerative Diseases, Issues, and Study Methodologies

Neurodegenerative diseases are chronic, incurable, and debilitating conditions originating from the progressive degeneration and/or death of neurons. Depending on the brain areas involved and on its connections, the neurodegenerative process can present with movement disorders and/or cognitive–behavioral diseases such as dementias [15].

Dementia is a clinical syndrome characterized by a progressive decline in cognitive functions associated with the loss of functional autonomy and behavioral–psychiatric symptoms, the etiology of which may depend on different diseases, both neurological and systemic [16]. Dementia affects about 50 million people worldwide and is related to dependence, poor quality of life, early institutionalization, and mortality [17]. Dementia has important social and economic implications both for direct medical and social care costs as well as for informal care. A global cost of dementia of 1.3 trillion USD was estimated in 2019, and this amount is expected to increase even more as both people with dementia and care costs are expected to increase in the coming years [18].

The etiology of most neurodegenerative dementias has been considered multifactorial, including both genetic and environmental factors. However, in many patients, the disease can be inherited as a Mendelian trait (i.e., monogenic form) [19]. Generally, neurodegenerative diseases involve some common pathogenic mechanisms, such as the accumulation, aggregation, or altered folding of proteins and/or mitochondrial dysfunctions that cause damage to the nervous system [20]. These pathological features lead to the manifestation of several symptoms that often overlap between different neurodegenerative diseases. Sharing clinical characteristics makes diagnosis particularly difficult [19]. Diagnosis is also made problematic by individual variability: patients with the same mutation in the same gene frequently manifest a range of different clinical symptoms [21], including variability in the age of onset [22]. Therefore, the use of a rigorous research methodology focusing on hereditary neurodegenerative diseases caused by a unique specific mutation can overcome the clinical heterogeneity, further permitting the building of large kindreds that are a useful model for neurodegeneration studies.

A complex research methodology—ranging from clinical activity to the genealogical reconstruction of families/populations across the centuries, the building of databases, and molecular/genetic research—was modelled on the characteristics of the Calabrian population. Specifically, the study starts from the clinical and diagnostic assessment of living patients that permits the families’ characterization. The genealogical reconstruction of the families/populations, in which the disease is transmitted, allows the identification of new affected surnames and consequently new patients with the same disease. Genealogical study requires the acquisition and analyses of birth, death, and marriage certificates in the municipalities (available from 1809) and of baptism, burial, marriage, and *Status Animarum* in the parishes (available from 1600). Data are further computerized in the e-database structure. The current database of the Regional Neurogenetic Centre (CRN) contains over 190,000 records of subjects linked by transitive relations, filiation, and marriage from different areas of Calabria [23,24].

## 3. Calabria as a Genetic Isolate for Neurodegenerative Diseases

Over a period of more than 25 years, using the methodology briefly summarized above, we identified many causative mutations of several neurodegenerative diseases that often manifest with atypical phenotypes (Table 1). These mutations, moreover, have shown a different distribution along Calabria, as reconstructed by the geographical localization of the patients followed at CRN (Figure 1).

In the following, we provide a description of the main research data collected on this topic.

### 3.1. Alzheimer’s Disease

Alzheimer’s disease (AD) is the most widespread type of dementia, clinically characterized by both cognitive and non-cognitive symptoms [40]. The main neuropathological features of AD are extracellular deposits of amyloid beta (Aβ) peptide and other molecules (amyloid plaques) and intracellular deposits of the hyperphosphorylated form of the protein tau (neurofibrillary tangles) [41]. AD can be classified as sporadic (SAD) and familial (FAD) based on the presence or absence of a clear hereditary genetic component [29,41]. In the familial forms, there has been documented mutations in three main genes: Presenilin 1 (PSEN1), Presenilin 2 (PSEN2), and Amyloid Beta Precursor Protein (APP), with an autosomal dominant inheritance (autosomal dominant AD, ADAD) [42].

Over the years, 1125 AD patients have been genetically tested at CRN. Among these, 93% (n = 975) did not show pathogenic variants and 7% (n = 150) were carriers of pathogenic variants (67% PSEN1, 27% APP, and 6% PSEN2) (*personal data*).

A clinical and genealogical study started in the 1980s allowed the reconstruction of one of the few very large families in the world, known as N family [43], which in 1995 was instrumental, together with another Italian family of Turin (TO family, [44]), in the identification and cloning of the PSEN1 gene, since both families carried the same Met146Leu mutation [45], thus demonstrated the same common origin.

Ancient clinical data for the N and TO families were found among medical records of the Psychiatric Hospital of Girifalco (Catanzaro) which operated from 1881 to 1978. It is worth mentioning the clinical record of Angela R., an ancestor of the N family—datable to 1904, before Alois Alzheimer’s description of the first case of this disease—which showed a clinical picture consistent with a diagnosis of non-amnestic probable AD, matching the ‘‘dysexecutive’’ phenotype described in her descendants. The a posteriori diagnosis of AD was supported by the evidence of the causative genetic mutation PSEN1 Met146Leu as well as neuropathological AD features in her genealogically proven descendants [46].

Our studies demonstrated that the PSEN1 Met146Leu mutation was a private and founder mutation in the Calabrian population shared among hundreds of affected subjects dispersed across several centuries and on several continents due to Calabrian emigration flow. The N-TO families are still considered a unique ADAD population reconstructed from the present time back to the 17th century over 11 generations and consisting of over 160 affected subjects who share a common ancestor calculated around the year 1000 [5]. The genealogical database reconstructed around the Calabrian kindreds contains subjects linked through the transitive filiation–marriage relationships and approximately 50,000 persons from 1600 to the present day [23]. Although these patients shared the same PSEN1 Met146Leu mutation, four different clinical pictures were identified: two classics for AD (memory deficits and spatial and temporal disorientation) and two characterized by symptoms pointing to frontal lobe involvement (apathetic and dysexecutive subgroups) [5].

This variability could be due to the localization of the early AD lesions in different cortical areas or to different unknown genetic modifiers [5] and underlines the broad phenotypic spectrum of Calabrian founder mutations.

In addition, the research on the N Family led to the identification of Nicastrin, a transmembrane glycoprotein that is part of the gamma secretase protein complex, which is one of the proteases involved in processing amyloid precursor protein (APP) to form the Aβ peptide [47]. The protein was so called to acknowledge the importance of the N Family, originating from the Calabrian village named Nicastro (today part of Lamezia Terme) [48].

Moreover, the characterization of other Calabrian families and subjects also allowed us to identify other novel mutations or variants associated with AD, such as *PSEN1* I143V [25], M84V [26], E318G [27], *PSEN2* Val139Met [27], and Ser130Leu [28].

Another important result, which reinforces the view of Calabria as a genetic isolate for neurodegenerative diseases, is related to the APP_A713T_ mutation. APP_A713T_ was initially described by Armstrong et al. [39] as a rare polymorphism with dominant inheritance associated with AD. However, from a first study that we carried out in 2004, there emerged an association of this mutation with early-onset dementia with multiple strokes in several members of a family from a village in central Calabria (PEC family). The neuropathological study of the proband revealed the presence of AD with severe cerebral amyloid angiopathy (CAA) and multiple infarcts [12]. Considering that other studies also revealed that some APP mutations (i.e., p.Ala692Gly, p.Glu693Lys, and p.Asp694Asn) can be associated with CAA [49], we decided to investigate the presence of APP_A713T_ mutation in 59 Calabrian patients affected by early-onset AD with cerebrovascular lesions (CVLs), a family history of dementia, and neuroradiological evidence of white matter lesions (WMLs) or hypodensities. The results of this study showed the presence of the APP_A713T_ mutation in heterozygosis in three late-onset unrelated patients living in different areas of the Calabria region (i.e., prevalence of 5%) [30]. The same mutation in homozygous affected subjects was also identified in another unrelated Calabrian family from a village in central Calabria affected by autosomal dominant AD with CVLs due to CAA [31]. Interestingly, both heterozygous and homozygous cases showed a similar clinical picture characterized by memory loss, absence of insight, and behavioral and personality changes [30,31]. More recently, we have reported the same mutation in a Belgian AD patient of presumed Italian descent and in another AD patient identified in northern Italy with Calabrian origin. Thus, we used a population genomics approach to estimate the inheritance from a common ancestor of the APP_A713T_ mutation in the Belgian and northern Italian patients and in six patients that were representative of all the apparently unrelated APP_A713T_ Calabrian families. The results showed that all carriers fell into the genetic variability of southern Italy. In addition, five out of seven patients shared a 1.7 Mbp-long DNA segment centered on the APP_713T_ mutation, making it possible to estimate the time of the most recent common ancestor for its common origin in the Calabrian region dating back over 1000 years [29].

Despite this evidence, in the Alzforum database the APP_A713T_ mutation is considered of uncertain significance [50]. However, the bioinformatical analysis on varsome [51] showed that it can be classified as “likely pathogenetic” according to the American College of Medical Genetics (ACMG) criteria [51]: (PM1) is located in a mutational hot spot and/or critical and well-established functional domain; (PP3) multiple lines of computational evidence support a deleterious effect on the gene or gene product; (PP5) a reputable source has recently reported this variant as pathogenic, but the evidence is not available to the laboratory to perform an independent evaluation. Further studies are thus needed to fill the knowledge gap in APP_A713T_ mutation and pathogenicity, at least in the Calabrian population.

### 3.2. Frontotemporal Dementia

Frontotemporal dementia (FTD) is the most prevalent neurodegenerative disorder with a presenile onset [52]. Based on clinical presentation, it can be classified into: (i) behavioral variant FTD (bvFTD); (ii) non-fluent primary progressive aphasia (PPA); and (iii) semantic variant primary progressive aphasia (svPPA) [53]. Recent estimates indicate that most cases of FTD are sporadic (55–75%), whereas the remaining 25–55% of cases are familial, in which mutations in three main genes have been documented: chromosome 9 open reading frame 72 (*C9ORF72*), microtubule associated protein tau (*MAPT*), and progranulin (*GRN*) [54]. These mutations give rise to a high heterogeneity of clinical and neuropathological manifestations with varying degrees of frontal and temporal lobe neuronal loss, atrophy, gliosis, and protein accumulation [55]. Specifically, *MAPT* mutations mainly determine the deposition of the microtubule-associated protein tau, whereas *C9ORF72* mutations determine the transactive response (TAR) DNA-binding protein of 43 kDa (TDP43). *GRN* mutations give rise to a predominance of diffuse granular neuronal cytoplasmic inclusions (NCIs) [56]. Many other mutated genes have also been associated with frontal dementia (e.g., *VCP, CHMP2B, TARDBP, FUS, SQSTM1, CHCHD10, TBK1, OPTN, CCNF, TIA1*), enlarging the genetic and phenotypic spectrum of this disease [57]. Over the years, 455 FTD patients have been genetically tested at CRN. Between these, 87% (n = 333) did not show pathogenic variants and 13% (n = 122) were carriers of pathogenic variants (39% *GRN*, 38% *C9ORF72*, 18% *MAPT*, and 5% *VCP*) (*personal data*).

In 2002, applying the same methodology previously adopted for AD, we reconstructed a large pedigree—known as B family—for 15 generations back to the 16th century in the village of Bivongi (Reggio Calabria, Italy). The corresponding database encompasses about 8000 persons from both the affected lineages and the unaffected (spousal) lineages. Thirty-four persons (19 females) over four consecutive generations have been identified as affected by FTD. All FTD patients have been linked to the same ancestors who lived in the early 18th century. Interestingly, although an autosomal dominant transmission was evident, none of the affected individuals had mutations of the *MAPT* gene—the only gene identified at that time as being associated with FTD [58]. In a subsequent door-to-door FTD study, we targeted all subjects in the B village who were ≥50 years of age (n = 702) [10] and investigated the *GRN* gene, which was discovered to be associated with FTD in other countries [59]. In all, 509 individuals (72%, mean age 71.6 ± 11.1; 55.4% females) agreed to participate, 6% were deceased, 10% lived outside the country, and 12% refused to take part. Between the participants, 23 belonged to B family. Of the group of 92 individuals (18.1%) evaluated in phase II (1 year later), 30 subjects were diagnosed with dementia (age 77.6 ± 5.6; 12 males/17 females) with a prevalence (number of cases per 100 individuals) of 5.9% among those ≥50 years of age and of 8.1% for those ≥65 years of age. Surprisingly, we found a very high and unusual prevalence rate (number of cases per 100 individuals older than age 49) of FTD (3.5%), followed by unspecified dementia (1.2%), AD (0.6)%, vascular dementia (VD) (0.4%), and Parkinson dementia (0.2%). In addition, we identified three different and novel mutations associated with FTD: one truncating *GRN* mutation (c.1145insA) and A266P and C126W. All mutations were associated with a very variable age of onset (between 35 and 87 years) and also to three distinct phenotypes (behavioral, affective, and delirious type) (Table 1), underlining the high prevalence and clinical variability that characterize the neurodegenerative diseases in the Calabrian population [10].

The characterization of other Calabrian families and subjects also allowed us to identify other novel mutations or variants associated with FTD, *MAPT* Val75Ala [34], V363I [35], IVS10 + 4A > C and IVS9 − 15T > C [36], and *GRN* Cys139Arg [32] and Cys139Arg [33], and in genes not generally associated with FTD such as *PRNP* P39L [37], *PSEN2* Arg62His [34], and *PSEN1* Val412Ile [36] (Table 1).

More interestingly, we have recently identified a novel mutation (D395A) in the *VCP* gene associated with early-onset FTD in a Calabrian family of the central area, not detected in 150 healthy subjects [38]. Previous data reported that mutations in this gene cause a rare multisystem proteinopathy known as inclusion body myopathy (IBM) associated with Paget’s disease of bone (PDB) and early-onset FTD (IBMPFD) [60]. *VCP* mutations have also been reported in patients with amyotrophic lateral sclerosis (ALS) [61], Charcot–Marie–Tooth type 2 (CMT2) disease [62], and hereditary spastic paraplegia (HSP) [63]. In addition, Saracino et al. [64] documented a *VCP*-mutated patient with FTD that did not develop clinical symptoms of PDB or IBM. In our case, the FTD was developed in three siblings without PDB and IBM signs, underlining the autosomal dominant transmission of this new mutation and, once again, the lack of a “classic” genotype–phenotype correlation of a mutation in members of the Calabrian population (Table 1). To our knowledge, our study offered the first description of the *VCP*-related FTD phenotype in patients belonging to the same family, suggesting that a *VCP* analysis should be considered for the genetic screening of familial FTD with an early-onset and in the absence of clinical signs of IBM or PDB, at least in the Calabrian population [38].

### 3.3. Parkinson’s Disease

Parkinson’s disease (PD) is the second most common neurodegenerative disease, with a prevalence of 1–2% in the population over 60 years of age [65], characterized by the presence of Lewy bodies and the loss of dopaminergic neurons in the pars compacta of the substantia nigra [66]. The main clinical features include both motor (e.g., bradykinesia, rigidity, postural instability, and resting tremor) and non-motor symptoms (e.g., hallucinations, anxiety, depression, cognitive dysfunction, sleep impairment, fatigue, and urinary disturbance) [67,68]. Although the etiology of PD is multifactorial, many causative genes have been identified, such as *α-synuclein*, *PARK2*, *PARK7*, *PINK1*, and *LRRK2* [69]. Between 2006 and 2011, we performed a screening to verify the presence of *LRRK2* mutations in 88 Calabrian patients affected by PD (63 sporadic and 25 with a family history of PD or other neurodegenerative diseases). Moreover, 200 motor and cognitively healthy control subjects (mean age 58.6 ± 12.3) were screened for each of the putative pathogenic sequence variants observed in PD cases [11]. In our cohort of patients, we reported a prevalence of 10.2% of *LRRK2* mutations, which appears to be higher compared to previous epidemiological studies on another population [70] and was not detected in the control subjects. In the same way, the frequency of the most common worldwide mutation, p.Gly2019Ser, was 3.2% higher in our sporadic cohort with respect to previously reported data [71], and it could be explained by the different ethnicity and the relative genetic isolation of the Calabrian population. In addition, this study allowed us to identify three novel missense variations of the *LRRK2* gene (p.Phe1227Leu, p.Gly1520Ala, and p.Ile2020Ser) associated with PD in Calabrian patients, underlying, once again, the peculiar genetic characteristics of this population and providing additional genetic insight into PD [11] (Table 1).

In recent decades, another research group has tried to characterize the genetics of PD in the Calabrian population, showing its peculiarities. For example, in Calabrian patients with PD, there has been reported a lower frequency of mutations in several genes, such as *CHCHD2* [72], *TARDBP* [73], *LRP10* [74], and *DNAJC13* [75], which instead are common in another cohort of PD patients [76,77,78,79]. Conversely, Calabrian PD patients’ data showed a higher prevalence of mutation of the *GBA* gene [80]. This evidence further strengthens the view of Calabria as a genetic isolate for neurodegenerative diseases and suggests continuing the search for novel mutations for PD in this population.

### 3.4. Niemann–Pick Type C Disease

Niemann–Pick type C disease (NPC) is a rare autosomal recessive inherited lipid storage disease caused by a defect in the intracellular trafficking of cholesterol [81]. The genes responsible for NPC are the *NPC1* gene, located on chromosome 18 and mutated in 95% of patients, and the *NPC2* gene located on chromosome 14 [82]. These genes encode for the NPC1 and NPC2 proteins that cooperatively mediate the egress of cholesterol from endosomes/lysosomes [83]. Due to the high heterogeneity of the clinical presentation, the disease is classified into five different forms based on the age of onset: perinatal (<2 months), early infantile (<2 years), infantile (<5 years), juvenile (<15 years), and adults (15–70 years) [84]. In adults, NPC often presents with a slowly worsening evolution typical of chronic neurodegenerative diseases.

Until a few years ago, the heterozygous status for *NPC1* or *NPC2* mutations was considered non-pathogenic, and heterozygous carriers were not supposed to develop any neurological symptoms during their lives. However, in 2013 a prevalence study [85] revealed a frequency of 3.6% for heterozygous mutations in *NPC* genes in a population of adults affected by dementia, parkinsonism, or psychosis. These data prompted us to speculate that, depending on the type of mutation, the heterozygous condition may induce an alteration in lipid metabolism and therefore a “benign” phenotype of NPC, with onset in adulthood and old age and a slowly progressive course, when the structure of the protein is altered. When the mutations do not alter the protein structure, a heterozygous status could be a risk factor for neurodegenerative diseases. To verify this hypothesis, we conducted a screening study for *NPC1* and *NPC2* mutations in 50 Calabrian patients affected by dementia with atypical clinical presentations or dementia plus, in which progressive and invalidating cognitive impairment was the main clinical feature associated with other neuropsychiatric and systemic symptoms [14]. Sequencing analysis revealed four different and known heterozygous mutations in *NPC1* (p. F284LfsX26 and c.1947 + 8G > C) and *NPC2* (p.V30M and c.441 + 1G > A) genes. The p.F284LfsX26 mutations were associated with a picture of progressive supranuclear palsy-like syndrome and the other three with a corticobasal syndrome (Table 1). The results of our study demonstrated that heterozygous mutations of *NPC1* and *NPC2* genes could contribute to dementia plus, at least in a subset of Calabrian patients.

Thus, heterozygosity can be a risk factor for dementia, and this is also confirmed by the high risk of neurodegenerative diseases, especially AD, in the parents of NCP patients [86] and by the links between lipid metabolism and Aβ [87]. Therefore, even the study of very rare forms can help to identify pathogenetic pathways of neurodegeneration that have not yet been fully elucidated.

### 3.5. Spinocerebellar Ataxia Type 17

Autosomal dominant cerebellar ataxia encompasses a group of neurodegenerative diseases clinically characterized by ataxia, ophthalmoplegia, pyramidal and extrapyramidal signs, and peripheral neuropathy. Dementia occurs only in some forms of spinocerebellar ataxia (SCA), such as SCA1, SCA2, SCA3, and SCA12, developing in the latest stages of the disease, while in SCA17 dementia is a constant feature of the phenotype [88]. A CAG repeat expansion in the TATA-box-binding protein (*TBP*) gene on chromosome 6 has been identified as the cause of SCA17 in some familial and sporadic cases, resulting in cerebellar ataxia and followed by dementia, parkinsonism, and dystonia, with onset in childhood and adulthood [89,90].

Starting from the end of the 1990s, our clinical and research attention was attracted by a large autosomal dominant Calabrian family with a complex neurologic syndrome that comprised early-onset dementia, psychotic features, extrapyramidal and cerebellar signs, and epilepsy [91]. The genealogic reconstruction of this family initially included 57 individuals (14 affected, 7 personally observed) across five generations. Since the clinical picture seemed to mimic different forms of neurodegenerative diseases in an atypical way, we decided to carry out on this family a large linkage analysis on 26 genes which were known to cause hereditary dementias (i.e., *APP*, *PSEN1*, *PSEN2*, *FTDP-17*, *BRI*, *PI12*, *FND*, *HD-like*, *SCA1*, *SCA2*, *SCA3*, *SCA4*, *SCA5*, *SCA6*, *SCA7*, *SCA8*, *SCA10*, *SCA11*, *SCA12*, *SCA13*, *PARK1*, *PARK2*, *PARK3*, *HD*, *DRPLA*, *PRNP*). Surprisingly, the molecular analyses had excluded the presence of mutations in these genes [91]. In subsequent years, we continued to reconstruct the genealogy of this family, including a total of 230 members across five generations, among whom 16 individuals were affected (4 men, 12 women; 11 personally observed). The observation of further cases allowed us to better characterize the clinical picture, namely by early and prominent behavioral disorders that, together with the strong reduction in verbal fluency, was followed by a definite picture of frontal lobe dementia. However, cerebellar signs were also noticed later but were eventually masked by extrapyramidal signs such as dystonia and rigidity. Myoclonus and epilepsy were characteristic of the late stages of the disease. The main neuropathological characteristics of the autopsied case were a low brain weight; atrophy of the frontotemporal cortex; nerve cell loss in the precentral gyrus, the primary visual cortex, the striatum, and the thalamic dorsomedial nucleus; pseudohypertrophic degeneration of the inferior olive; and severe loss of Purkinje cells. This evidence led us to hypothesize that it could be an atypical clinical picture of SCA17. The molecular analysis performed on the *TBP* gene confirmed our hypothesis. Specifically, we found a stable CAG repeat expansion in this gene, despite the reported differences in the age of onset among generations three, four, and five (from 17 to 53) [13] (Table 1). Thus, the characteristics of this Calabrian family broaden the clinical picture of SCA17: initial presenile dementia with behavioral symptoms should be added to ataxia, rigidity, and dystonic movements, which are more commonly encountered.

## 4. Discussion

Neurodegenerative diseases are highly variable and heterogeneous concerning their causes and phenotypes [92]. Being, moreover, age-related diseases, metabolic and vascular risk factors are frequently co-occurring, thus increasing confusion and complicating clinical and pathological aspects. The clinical and genetic research conducted on neurodegenerative diseases can simplify and reduce the variability by using a “simple model” that is constituted by the large families/kindreds and populations in which neurodegenerative mendelian diseases segregate. The genotype–phenotype correlation studies have largely improved the clinical knowledge of these diseases, and the relatively recent epigenetic and epigenomic research has addressed the undeniable importance of precision medicine [93].

Despite the enormous advancement of knowledge concerning the etiology and mechanisms of pathogenesis, much is still obscure, and to date there is no cure stopping or modifying the fatal course of neurodegenerative disorders. However, the enormous socioeconomic impact of these diseases due to the global aging of populations makes it more and more urgent to better clarify mechanisms, identify all links in the chains, and possibly find a cure (or many) [94].

Genetic isolates are characterized by geographical and cultural isolation and low genetic variability due to a lack of immigration and, consequently, high inbreeding. These populations offer an advantage in finely characterizing the genetic architecture of complex disorders due to the high frequency of the common pathological trait(s) and, therefore, the possibility of reconstructing genealogical family trees, building large kindreds, thus allowing the possibility of tracing back mutations across the centuries and rather easily conducting wide-ranging molecular genetics analysis to isolate causative mutations in pathological genes.

In the rest of the world, only a few places have been described as genetic isolates and made it possible to achieve important results on several diseases [95]. An Italian example is Sardinia, where founder mutations have been described in several medical areas, including neurological disorders such as Wilson’s disease [96], amyotrophic lateral sclerosis [97], and Parkinson’s disease [98]. In the same manner, in Sicily there has been reported a high prevalence of certain diseases, such as hereditary amyloid transthyretin (ATTRv) polyneuropathy [99].

The clinical and research work carried out over approximately thirty years in Calabria and developed with several research groups around the world, demonstrated the validity of the model for all the neurodegenerative diseases studied and undoubtedly increased the advancement of knowledge in this field. The study on autosomal dominant Alzheimer’s disease (ADAD) mainly contributed to the isolation and cloning of the *PSEN1* gene in 1995 [45], the identification of the novel gamma secretase component protein “Nicastrin” [47], the amyloid cascade hypothesis, and the description of one of the largest families carrying ADAD in the world [5]. The genotype–phenotype correlation studies on ADAD patients have provided the possibility to better characterize early phenotypes showing frontal involvement or other atypical presentation, and these clinical specific aspects were only successfully recognized by the NIA–AA criteria, for the AD diagnosis, in 2011 [100]. Clinical follow-up of Dominantly Inherited Alzheimer Network (DIAN) cohorts showed that the biological disease starts in brains of subjects carrying mutations approximatively 20 to 25 years before the clinical onset of the disease [101].

FTD in the Biv. area of Calabria shows a prevalence notably higher (3.5%) compared to AD (0.6%) and no report exists to date in other countries of the world or other Italian regions. For example, according to worldwide data [102], it has been reported that AD is the most prevalent form of dementia in a mountainous village in Sicily [103] [CIT], Brescia County and Vescovato (Lombardia Region) [104,105], Macerata County (Marche Region) [106], Verona (Veneto Region) [107], and Vecchiano (Toscana Region) [108].

In the Biv. area, certain causes of this high prevalence of FTD have been identified in three different *GRN* mutations [10], but numerous causes remain to be determined and the etiology and epigenetic aspects need to be studied. Peculiar environments rich in metal mines could be a trace to follow in the future.

Notably, we have also reported a higher prevalence of PD associated with *LRRK2* mutations with a broad range of onset (51–76 years) in Calabria [11] compared to the Italian population [109], even in this case probably due to the Calabrian geographical and historical isolation that has occurred in the last three centuries [11]. This latter aspect is also confirmed by the heterozygous mutations of *NPC1* and *NPC2* genes that could contribute to dementia plus, at least in a subset of Calabrian patients [14], and by the novel *TBP* mutations that we found associated with an atypical clinical picture of SCA17 in a large autosomal dominant Calabrian family [9].

Moreover, almost all the neurodegenerative diseases studied in Calabria show peculiarities. They are frequently associated with many novel causative or associated mutations or variants that often manifest with atypical phenotypes compared to the “classical” clinical pictures and at a younger age. These data contributed to the results of a recent metanalysis that reported a worldwide higher prevalence of early-onset dementia (119 per 100,000 inhabitants) [110].

Often, in early-onset Calabrian patients, the cognitive impairment is accompanied by a higher frequency of neuropsychiatric symptoms, such as apathy, agitation, depression, hallucinations, anxiety, disinhibition, and eating disorders [111]. The cause for increased neuropsychiatric symptoms in these patients is likely multifactorial and includes both social and biological factors.

Although the different genetic backgrounds of genetic isolate areas has been generally considered to have relatively little impact due to the rarity of the “rare diseases” [112], this does not apply to the Calabrian population. Knowing the peculiarities of Calabria as a genetic isolate and its neurodegenerative diseases can be important for the international medical and scientific community as the emigration to different places in Italy and the world (e.g., Australia, North and South America, Canada, Europe, England, etc.) occurred during the last few centuries [113] and especially in recent years [114]. Notably, it was estimated that in the world there are about 7 million first-generation and 30 million third-generation Calabrians [115] out of the 1,839,352 currently living in the region [1]. Thus, emigration constantly brings millions of people out of this region, and individuals carrying rare mutations with their potential diseases risk the difficulty of being diagnosed out of their geographically informed context.

In addition, we must remember that we are facing dominant diseases whose onset is in adulthood when the subjects, in most cases, already have children. Receiving a diagnosis of a neurodegenerative disease is surely more emotionally difficult for these patients due to their younger age and greater responsibilities within their families such as taking care of children and holding down a job [116]. Moreover, the high variability in age of onset and phenotype that we have documented at the level of the same mutation (e.g., APP_A7113T_) makes it desirable that a genetic screening will be carried out on all patients presenting symptoms of neurodegenerative diseases in Calabria, especially if they manifest atypical clinical pictures and have a positive family history of these disorders. This could not only allow us to identify other patients who present the mutations that we have already described but also further new ones not previously reported in the literature, opening new ways of both pathologically characterizing neurodegenerative diseases and developing new treatments. Moreover, all these specific aspects suggest that health services should be built for young onset forms of dementia (e.g., family counseling services and social services). General practitioners also need to be trained in all these diseases for an early diagnosis and intervention. The community must also be informed and made aware of the existence of the early-onset forms that require not only medical and social care but also a reduction of social stigma to improve the quality of life of both patients and their families. Last but not least, these patients (and also at-risk subjects) could accept experimental clinical trials such as of the novel disease-modifying drugs, as already done in prevention trials kicked off by DIAN-TU [117,118].

## 5. Conclusions

In conclusion, the Calabrian population could represent a useful model not only for the characterization of the etiology and pathophysiology of several debilitating and currently incurable neurodegenerative diseases but also for the development of new treatments suitable all over the world.

## Figures and Tables

**Figure 1 biomedicines-10-02288-f001:**
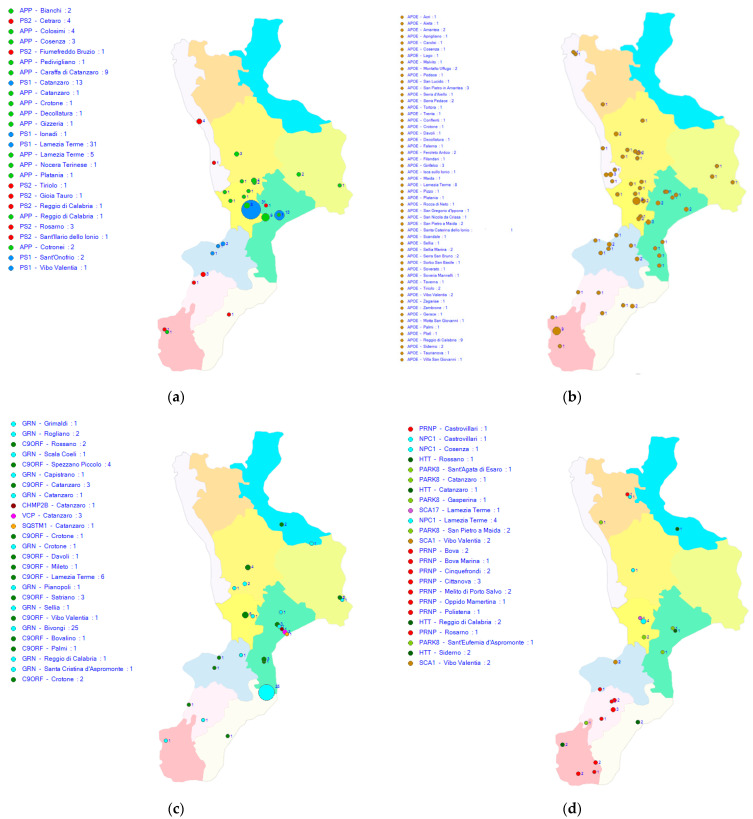
Geographical distribution of mutations in Calabria: (**a**) mutations associated with Alzheimer’s disease; (**b**) APOEε4 carriers affected by Alzheimer’s disease; (**c**) mutations associated with frontotemporal dementia and amyotrophic lateral sclerosis; (**d**) mutation associated with other neurodegenerative diseases (unpublished data).

**Table 1 biomedicines-10-02288-t001:** Novel mutations or variants associated with neurodegenerative diseases in Calabrian population.

Neurodegenerative Disease	Mutation or Variant	Patient(s)	Clinical Features	Neuroimaging/Neuropathology	Reference
** *Alzheimer’s Disease* **	*PSEN1* I143V	Female proband of a four-generation family with history of dementia (11 subjects: 5 females, 6 males). Two more subjects were reported as affected by depression.	Onset at 55 years with personality changes, apathy, reduction of verbal fluency, and temporal and spatial disorientation. At 68, appearance of visual hallucinations, slurred speech, and rigidity. She became bedridden and died at 75.	The brain showed severe atrophy of the frontal and temporal lobes. Parenchymal amyloid-β (Aβ) deposits were abundant, diffuse grey structures, and contained Aβ42, but very little Aβ40. Amyloid angiopathy was absent. Neurofibrillary changes were severe.	[25]
	*PSEN1* M84V	Three patients belonging to a large kindred affected by ADAD (14 over three generations).	Early-onset (mean age = 52.7 years), frontal executive syndrome, psychosis and spastic paraparesis.	Diffuse cortical and cerebellar atrophy.	[26]
	*PSEN1* E318G and *PSEN2* Val139Met	One patient affected by late-onset familial AD. The father had died at about 80 years old with a picture of dementia clinically resembling that of the proband, and two sisters had died at 83 and 74 years old, both showing a progressive picture of dementia starting with memory impairment, cognitive deficits, time and space disorientation, and agitation evolving into a severe state of dementia.	Proband was 76 years old when he presented with memory and language deficits. Symptoms progressed very slowly together with episodes of spatial disorientation; he frequently showed mistakes in identifying relatives and friends and became temporally disoriented and unable to perform activities of daily living. Irritability and agitation were noticed by relatives, but only when the behavioral disorder worsened did the family ask for a neurological consultation, five years after the onset. Neurological examination was performed and found normal when the patient was 83; he exhibited moderate temporal and spatial disorientation, showing a selective and moderate impairment of verbal long-term memory and of abstract thinking. At 86, he showed dysphasic and incomprehensible speech, was unable to recognize children, and unable to walk alone. He was still alive but bedridden.	Bilateral hypoperfusion at the parietal–temporal lobes more marked in the right hemisphere.	[27]
	*PSEN2* Ser130Leu	One male patient with a familial history of two first-degree relatives with a PD without cognitive impairment.	Late-onset AD. At 81 years old, his wife noted progressive memory impairment and episodes of spatial disorientation while driving. His mood and personality changed. He started to shun social contact, losing interest in friends and in leisure activities; he developed delusions. On examination, the patient was depressed, apathetic, and evidenced moderate impairment of cognition. Neurological examination was normal. Six months later, the patient showed worsened memory functions. He rapidly developed language deficits, agitation, delusion, and hallucinations. One year later, the patient was amnesic, agnosic, aphasic, and showed mild bradykinesia.	Mild temporal atrophy without vascular lesions.	[28]
	*APP* A713T *	PEC family (21 subjects: 13 males, 8 females) over the last three generations were identified as at risk or affected by dementia; 8 persons were reported to be affected by dementia through history (5 males, 3 females), and 5 were studied clinically and genetically (5 males); 8 asymptomatic at-risk subjects (3 males, 5 females) were also clinically and genetically investigated. Another two patients were identified in northern Italy and in the Walloon region of Belgium. All subjects shared a common ancestor who lived in Calabria 1000 years ago.	Early- or late-onset AD with severe cerebral amyloid angiopathy (CAA) and cerebrovascular lesions (CVLs), both in heterozygosis and in homozygosis.	Severe atrophy in parietotemporal cortices or frontoparietotemporal cortices and cerebrovascular lesions in the cortex with frontoparietotemporal hypoperfusion.	[12,29,30,31]
** *Frontotemporal dementia* **	*GRN* c.1145insA, A266P, and C126W	Twelve carriers of the c.1145insA, three carriers of the A266P, belonging and segregating in two different branches of the same large kindred (B-family). Two apparently sporadic patients, also of Bivongi, descending from a common founder identified in the sixth generation.	A difference in the age at onset was evident among the three groups (c.1145insA carriers (64.2 ± 12.5 years); A266P carriers (75.7 ± 2.9 years); C126W carriers (71 ± 8.5 years)). Three distinct FTD phenotypes correlated to each *GRN* mutation. Carriers of c.1145insA showed a more severe “frontal Gestalt” with marked distractibility, disinhibition, a greater impairment of social awareness, and language impairment developing into complete mutism (*behavioral type*). A266P carriers showed affective symptoms with apathy in 100% of cases, emotional unconcern, and verbal output reduction that remained stable for the entire course of the disease (*affective type*), while C126W carriers presented with paranoid delusions and agitation (*delirious type*).	Data not shown.	[10]
	*GRN* Cys139Arg	One male patient. The sister, deceased at age 70, was reportedly affected by a similar clinical picture of dementia.	At age 61, the patient presented with slow progressive personality changes and behavioral disturbances. He was playful and moriatic, unaware of his changes, and also displayed an alteration in eating behavior and motor stereotypes. He became unable to drive. Within 3 years indifference, apathy, and reduction of verbal fluency appeared; memory functions in contrast were relatively preserved. At age 69, the patient was mute, incontinent, and completely dependent on others for care. However, there was no evidence of any clinical motor impairment.	Not available.	[32]
		Three patients coming from Calabria (patient CI2), Tuscany (patient TM1), and Lazio (patient LT3). CI2 had a family history of dementia and parkinsonism; TM1 had a family history of psychiatric disorders but not of dementia; LT3 had nine siblings, four of whom were affected by dementia without available DNA and blood samples.	The patients presented with different clinical pictures (TM1, bvFTD; CI2, semantic dementia, SD; and LT3, corticobasal degeneration, CBS) and showed a wide range of age at onset (55–80 years old), which confirmed the heterogeneity of clinical phenotypes associated with this mutation.	*CI2*: atrophy and hypometabolism of the left temporopolar and temporoparietal areas. *TM1*: cortical frontal atrophy associated with subcortical white matter abnormalities and hypometabolism in the bilateral parietotemporal and left frontal dorsolateral areas. *LT3:* marked atrophy of both frontal and temporal anterior lobes, asymmetric for left side, and a reduced perfusion in superior and medial frontal gyri and in temporoparietal areas.	[33]
	*MAPT* Val75Ala and *PSEN2* Arg62His ^#^	Two siblings; a male carrying the *MAPT* Val75Ala mutation and a female carrying the *PSEN2* Arg62His mutation. Their mother presented with a mood disorder; their father died at 66 years from myocardial infarction; two more brothers were healthy. A paternal aunt and the paternal grandmother were affected by AD and PD, respectively.	The clinical picture shown by the affected subjects was consistent with early-onset FTD. Both siblings presented with a similar behavioral pattern of dementia characterized by insidious onset, disinhibition, loss of social awareness, and absence of insight. Impairment of executive functions evidenced involvement of the frontal lobe together with phonological verbal fluency which was severely impaired in both patients. They also shared perseveration and utilization behavior that are typical signs of frontal involvement. Memory functions were compromised at different levels due to the different stages of the disease; constructional apraxia was relatively spared in both. The patients were unaware of their pervasive changes. Both patients showed myoclonus and developed epileptic seizures. Through molecular analysis, we excluded other dementing disorders featuring myoclonus and epilepsy, such as familial AD and CJD.	The male patient showed a frontotemporal atrophy and frontomesial and frontoparietal left hypoperfusion. Theta and delta activities were present at the EEG in frontotemporal areas. The female patient showed a moderate cortical and subcortical atrophy and a marked hypoperfusion in frontal lobes prevalent on the left hemisphere. EEG showed diffuse slow waves.	[34]
	*MAPT* V363I	One female proband of a three-generation family with four other subjects who were carriers of the same mutation asymptomatically. However, only the proband was homozygous for the A allele of *PGRN* SNP rs9897526 and for methionine at 129 codon of the *PRNP* gene, suggesting that the genotype of some modifier dementia genes might contribute to the pathogenic effect of the *MAPT* V363I variation.	The proband was 53 years old when, after an infectious episode, she began to manifest visual hallucinations and progressive personality changes. She became disinhibited with inappropriate jocularity; she touched everything and everyone and stopped people in the street to say that she was seeing angels. Insight was absent. She danced all day, listened to music, drew on the walls, and refused cooking and home management. In contrast with an excessive use of make-up, she neglected personal hygiene. She became totally unable to work and began treatment with haloperidol. Apathy, emotional blunting, reduction of verbal initiative, early incontinence, and inability to perform daily living activities appeared. No improvement arose when the treatment was stopped. At the age of 56 years, the patient was excessively smiling, echolalic, answered only simple questions, sang old songs, but continued to dress correctly. She was untestable; both pyramidal and supraspinal signs were present. Although symptoms gradually progressed to complete mutism and hyperorality, she continued to manage known environments. The patient was clinically diagnosed with FTD. Within 3 years, the patient became amimic, showing myoclonus, and presenting with massive flexor rigidity of the neck and both arms; walking became impossible. She died at the age of 61 years.	Massive cortical and subcortical atrophy.	[35]
	*MAPT* IVS9 − 15T > C and IVS10 + 4A > C	One female proband. The IVS10 + 4A > C mutation was present in the proband’s mother and in four maternal uncles. The IVS915T > C mutation was present in the proband’s father and one of her sisters, while only one sister had both mutations and showed slight verbal memory disturbances together with mild deficit in attention and concentration. Both the proband and her sister with both mutations were heterozygous for MAPT H1/H2; APOE genotypes were, respectively, ε2/ε3 and ε3/ε3, and the M129V polymorphism of the *PRNP* gene was MM in the proband and VV in her sister.	The proband shown presenile dementia with a clinical picture characterized by apathy, depression, and absence of insight, followed by pyramidal and extrapyramidal signs consistent with FTD.	Autopsy of the proband’s brain revealed a massive and selective atrophy of the frontal and temporal lobes, severe atrophy of the caudate nucleus and of the white matter; the hippocampus was destroyed. An exceptional neuronal loss in all layers of the frontotemporal and insular cortices with intense protoplasmatic gliosis were present. Abundant neurofibrillary tangles and Pick body-like inclusions in neurons, tufted astrocytes, and coiled bodies in oligodendrocytes, consistent with an increased expression of 3R tau.	[36]
	*PRNP* P39L	*Patient 1*: male with four other family members affected by behavioral disturbances and cognitive impairment suggestive of FTD. *Patient 2*: Male with no family history suggestive of dementia, although his mother was referred to as an irritable and willful person. She was impulsive and bossy like the patient and in the later years of her life showed signs of hand tremor.	Both patients shared a frontal dementia dominated by a dysexecutive syndrome and severe behavioral disturbances, whereas psychotic signs such as delusions or hallucinations were absent. Neither patient developed any of the typical clinical signs associated with prion diseases, such as myoclonus or cerebellar ataxia.	*Patient 1:* Cortical atrophy with a marked involvement of mesial, frontal, temporal, and posterior parietal regions in the left side, besides some lacunar ischemic subcortical lesions. Hypoperfusion of frontotemporoparietal cortical areas with relative sparing of the occipital lobe, brainstem, and cerebellum. EEG did not show any typical periodic complexes of prion diseases. *Patient 2:* Diffuse cortical atrophy prominent in mesial frontal, temporal, and posterior parietal regions, mainly in the left side. EMG and EEG were normal.	[37]
	*PSEN1* Val412Ile	FUS family of four affected subjects in two generations.	The patients shared a similar clinical picture, which included behavioral disorder and deficits on neuropsychological tests, fulfilling the clinical criteria for FTD, despite the AD expected phenotype caused by *PSEN1* mutations.	Two patients presented diffuse cortical hypometabolism; the other two were unavailable for neuroradiological examinations.	[32]
	*VCP* D395A	Three siblings (two sisters; one brother) with a possible family history of dementia (the father died in a psychiatric hospital at 60 years old).	Early-onset behavioral variant of FTD without inclusion body myopathy (IBM) and Paget’s disease of bone (PDB), unlike many reports on *VCP* disease.	*Patient 1:* diffuse cortical atrophy prominent in frontal and temporal regions and hypoperfusion in frontal and temporal convolutions of the left hemisphere and upper frontal circumvolution of the right hemisphere. *Patient 2:* diffuse cortical atrophy prominent in the frontal lobes and bilateral frontoparietotemporal hypoperfusion. *Patient 3*: diffuse cortical atrophy and bilateral frontoparietotemporal hypoperfusion.	[38]
** *Parkinson’s disease* **	*LRRK2* p.Phe1227Leu	One female patient with family occurrence of Parkinson’s disease.	Onset at 65 years with bradykinesia. The patient also developed tremor at rest and rigidity accompanied by depression.	Data not shown.	
	*LRRK2* p.Gly1520Ala	One male patient with family occurrence of Parkinson’s disease.	Onset at 68 years with bradykinesia and freezing of gait. The patient also developed tremor at rest, postural tremor, rigidity, and postural instability.	Data not shown.	[11]
	*LRRK2* p.Ile2020Ser	One male patient without family occurrence of Parkinson’s disease.	Onset at 51 years with bradykinesia. The patient also developed tremor at rest, rigidity, dyskinesias, and postural instability, accompanied by depression.	Data not shown.	
** *Niemann–Pick type C Disease* ^+^ **	*NPC1* p. F284LfsX26	One female patient with a family history of late-onset dementia.	The 68-year-old patient, with moderate congenital mental retardation and poor acquisition of language and judgment abilities, was affected by a progressive supranuclear palsy-like syndrome. The NPC-SI score was 116 indicating high suspicion of NPC. The onset was at 64 years, with speech, gait impairment, and postural limb tremor. In the following years, she developed progressive cognitive deterioration with prominent executive dysfunction and behavioral disturbances. She also showed loss of autonomous ambulation, anarthria, severe dysphagia, global bradykinesia with small-step gait, freezing, and postural instability, vertical gaze palsy, fixed oral and cranium–cervical dystonia with anterocollis, right focal limb dystonia, axial and right limb cogwheel rigidity, and ideomotor apraxia. Severe dementia was detected with apathy, sleep disturbances, and irritability. The patient also presented steatosis, hepatomegaly, and moderate cholesterol accumulation compatible with a variant biochemical phenotype. Plasma levels of cholestane-3β,5α,6β-triol were normal.	Severe midbrain atrophy (“hummingbird sign”) and marked cortical atrophy with reduced striatal uptakes in the left putamen.	[14]
	*NPC1* c.1947 + 8G > C	One female patient with family history of depression.	Patient was a 62-year-old with mild mental retardation, affected by corticobasal syndrome. The NPC-SI score was 97 indicating high suspicion of NPC. Onset at 32 years with depressive symptoms, evolving into apathy, emotional lability, and short-term memory impairment at 41 years. Both “absence-like” seizures and gelastic cataplexy with sudden weakness associated with strong emotions, particularly laughter, occurred. At 56 years, the patient appeared emotionally flat and apathetic, with moriatic and imitative behaviors, wandering, binge eating, and craving for sweets. She had insight of her symptoms, but she was emotionally indifferent and quite theatrical in reporting them. She became careless and incurred an ocular injury with a mishandled pen, losing sight in her right eye. In subsequent years, she developed frontal dementia, showing impaired abstract reasoning, attention, and awareness, as well as anomia and dressing apraxia. Clinical examination detected mild gait impairment and left limb hemidystonia associated with pyramidal syndrome, ideomotor apraxia, echolalia, and mirror movements. Vertical gaze was normal. Neuropsychological evaluation confirmed a prominent impairment of executive functions, abstract reasoning, and judgment abilities associated with environmental dependency. Abdominal echotomography showed slight hepatomegaly. Filipin staining was negative and plasmatic levels of cholestane-3α,5β,6α-triol were normal.	Severe frontal, parietal, and occipital cortical atrophy, mainly in the right side, and atrophy of the left temporal lobe and diffuse cortical hypoperfusion, mainly in the left temporoparietal region.	
	*NPC2* p.V30M	One female patient without family history of NPC.	Patient was 52 years old and affected by corticobasal syndrome. The NPC-SI score was 132 indicating high suspicion for NPC. Onset at 39 years with pain in the left shoulder and hand clumsiness that led to a reduced ability to perform housework. After three years, the patient showed planning and executive deficits and difficulties in word finding. She developed motor slowing and gait disturbances and was unresponsive to L-DOPA. Personality changed with apathy, fatuous and childish behavior, and emotional incontinence. By the age of 50 years, we observed progressive worsening of gait and the patient was constrained to a wheelchair, in addition to exhibiting speech disorder, dysphagia, severe frontal dementia, and visual hallucinations. Concurrently, she presented with myoclonic jerks, generalized epileptic seizures, and REM sleep behavior disorder. At first clinical examination, the patient showed a severe akinetic–rigid syndrome associated with gait apraxia and freezing, dysarthria and speech apraxia, vertical gaze palsy and oculomotor apraxia, focal limb, and trunk dystonia. Non-fluent aphasia and ideomotor apraxia were also detected. Filipin staining was inconclusive, showing a few cells with weak fluorescent perinuclear vesicles. Plasmatic levels of cholestane 3α,5β,6α-triol were normal.	Severe atrophy of the right frontal and parietal lobes and moderate atrophy of the cerebellar hemispheres and severe hypometabolism of the occipital and parietal cortical areas with a reduced striatal uptake in the right putamen.	
	*NPC2* c.441 + 1G > A	One male patient with a family history of cognitive impairment, psychiatric symptoms, and substance addiction.	A 70-year-old patient affected by corticobasal syndrome. The NPC-SI score was 111 indicating high suspicion of NPC. Onset at 61 years, with difficulties in word finding, personality changes, and inability to drive. At 67 years, he was incapable of oral expression and writing and experienced difficulties in using and recognizing common objects. Daily activities were progressively abandoned because of both impaired action planning and disrupted awareness of positions and spatial relationships of objects in the environment. Memory was not compromised. Nocturnal myoclonus was reported. At 68 years, visual hallucinations, confabulation, and delusions. Simultaneously, he developed an extrapyramidal syndrome with gait impairment, action tremor, and rigidity as well as global disuse of the right limbs, without improvement on withdrawal of antipsychotic drugs. Clinical examination at age 67 showed vertical gaze palsy, non-fluent aphasia, speech apraxia, ideomotor oculomotor and orofacial apraxia, and postural limb tremor. After three years, additions to the neurological picture included anarthria and right asymmetric akinetic–rigid parkinsonism with gait apraxia, dystonia, and myoclonic jerks of the right limbs. Cognitive evaluation confirmed non-fluent aphasia, agraphia, constructive apraxia, disorientation, and executive dysfunction. Behavioral assessment revealed persistent hallucinations, delusions, apathy, lability and agitation, anxiety, and wandering. Filipin staining revealed moderate intracellular cholesterol accumulation compatible with a “variant biochemical phenotype”. Plasma levels of cholestane-3α,5β,6α-triol showed normal values.	Cortical atrophy of the frontal and parietal areas and the left temporal lobe with asymmetrical cortical hypoperfusion in the left posterior associative areas.	
** *Spinocerebellar Ataxia Type 17* **	*TBP* CAG/CAA repeat expansion	Sixteen individuals affected (four men and twelve women) across five generations of a large kindred.	Initial presenile frontal-type dementia with behavioral symptoms, ataxia, rigidity, and dystonic movements.	Global atrophy of the cerebral and cerebellar cortices. The main neuropathological characteristics of the autopsied case were a low brain weight, atrophy of the frontotemporal cortex, nerve cell loss in the precentral gyrus, the primary visual cortex, the striatum, and the thalamic dorsomedial nucleus, pseudohypertrophic degeneration of the inferior olive, and severe loss of Purkinje cells.	[13]

*Note*. Some of these mutations or variants were previously described as non-pathogenic or associated with other diseases or in compound with other heterozygous mutations. * This mutation had already been reported as a rare polymorphism associated with AD [39]. ^#^ This mutation was previously reported as associated with AD [34]. **+** These mutations had already been identified as associated with NPC in compound heterozygosity with other mutations [14].

## Data Availability

The personal data presented in this study are available on request from the corresponding author. The data are not publicly available because preliminary.

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
