# Peer review of "Calabria as a Genetic Isolate: A Model for the Study of Neurodegenerative Diseases"

_biomedicines, 2022, doi:10.3390/biomedicines10092288_

Round 1
Reviewer 1 Report
This is an interesting paper describing molecular genetic testing in neurodegenerative diseases in Calabria. I think that this study is of interest and may arise some concerns about genetic testing in special regions with characteristic genetic background. The article is clear and very interesting, but I have some concerns and considerations:
1) I think that the paper may be consistently improved by a further analysis of the numbers needed to be tested in order to have a positive result by genetic testing. I suppose that this index can vary on the basis of the disease rarity, giving some practical points and feedback to clinicians who ask for genetic testing. For example, it may be of interest to know if it could be easy to find a diagnostic mutation in allelic disorders in parkinsonism rather than ataxia or dementia. I suggest to report percentages in graphs or tables in order to make data more useful for the reader.
2) It should be discussed the eventuality of finding new genetic variants, how it was common in your study and how it influenced the clinical practice; the authors discussed about new variants, but they only cited them; I suggest to report a table describing all new variants and their phenotypes. These data are very important and might significantly increase the interest for this review article adding original data.
3) C9ORF72: it is a well-known fact that this genetic alteration can be associated with many phenotypes; which phenotypes did you encountered (ataxia, autoimmune, ALS, dementia, ALS-dementia, …)? Report them in your experience and discuss.
4) LRKK2 mutations: 10% is very high! Add phenotypes associated with new mutation in a specific table as suggested before.
5) Lines 424-425. Another example is Sicily concerning hereditary amyloidosis (ATTRv). I think that a comparison with this special neighboring area is necessary. I suggest a reading on epidemiological data in ATTRv (https://pubmed.ncbi.nlm.nih.gov/33925301/).
6) Conclusion is elusive and too long. I suggest to move its parts to the introduction and discussion concentrating a brief message in just a paragraph (8-9 lines).
7) Grammar and style are adequate.
Author Response
Thank you very much for your timely and very useful review of our paper. We know that reviewing takes time, and we are thankful for the amendments. We have revised our manuscript based on your comments. We believe our manuscript is now more robust due to your comments.
In the following, we have answered each point raised by you.
Comment 1. I think that the paper may be consistently improved by a further analysis of the numbers needed to be tested in order to have a positive result by genetic testing. I suppose that this index can vary on the basis of the disease rarity, giving some practical points and feedback to clinicians who ask for genetic testing. For example, it may be of interest to know if it could be easy to find a diagnostic mutation in allelic disorders in parkinsonism rather than ataxia or dementia. I suggest to report percentages in graphs or tables in order to make data more useful for the reader.
Response 1. Thank you for this feedback. We have now added in the main text how many patients we genetically tested to identify those with mutations and the percentages of the different mutations we found. In addition we have now added the number of patients and control (and percentage) that we tested in each main study to establish the prevalence rate of a given mutation (marked in red).
Comment 2. It should be discussed the eventuality of finding new genetic variants, how it was common in your study and how it influenced the clinical practice; the authors discussed about new variants, but they only cited them; I suggest to report a table describing all new variants and their phenotypes. These data are very important and might significantly increase the interest for this review article adding original data.
Response 2. Thank you for this feedback. We have now discussed the eventuality of finding new genetic variants, how it was common in our study and how it influenced the clinical practice (marked in red in the discussion). We have also report in a table (table 1) a description of all new mutations or variant and their phenotype.
Comment 3. C9ORF72: it is a well-known fact that this genetic alteration can be associated with many phenotypes; which phenotypes did you encountered (ataxia, autoimmune, ALS, dementia, ALS-dementia, …)? Report them in your experience and discuss.
Response 3. Thank you for this feedback. We have only unpublished data on C9ORF mutation and we are now preparing a manuscript specific on this topic. However, we have now insert in the main text the number and percentage of FTD patients with this mutation (marked in red).
Comment 4. LRKK2 mutations: 10% is very high! Add phenotypes associated with new mutation in a specific table as suggested before.
Response 4. Thank you for this feedback. We have now added these information in table 1.
Comment 5. Lines 424-425. Another example is Sicily concerning hereditary amyloidosis (ATTRv). I think that a comparison with this special neighboring area is necessary. I suggest a reading on epidemiological data in ATTRv (https://pubmed.ncbi.nlm.nih.gov/33925301/).
Response 5. Thank you for this feedback. We have now added these information in the discussion (marked in red).
Comment 6. Conclusion is elusive and too long. I suggest to move its parts to the introduction and discussion concentrating a brief message in just a paragraph (8-9 lines).
Response 6. Thank you for this feedback. We have now divided the previous conclusion into discussion and conclusion.
Comment 7. Grammar and style are adequate.
Response 7. Thank you for this feedback.
Reviewer 2 Report
The authors summarized and discussed some research results that support the idea that Calabria is a good useful model for the study and characterization of neurodegenerative diseases from a genetic point of view. I believe the paper is publishable, but I suggest the authors provide comparative examples with other region/s in order to strengthen the data described.
Author Response
Comment. The authors summarized and discussed some research results that support the idea that Calabria is a good useful model for the study and characterization of neurodegenerative diseases from a genetic point of view. I believe the paper is publishable, but I suggest the authors provide comparative examples with other region/s in order to strengthen the data described.
Response. Thank you for this feedback. We have now added these information in the discussion (marked in red).
Reviewer 3 Report
The manuscript is well written and presented. It should be published.
Author Response
Thank you for this feedback!